# Composition and Fatty Acid Profile of Bone Marrow in Farmed Fallow Deer (*Dama dama*) Depending on Diet

**DOI:** 10.3390/ani12080941

**Published:** 2022-04-07

**Authors:** Żaneta Steiner-Bogdaszewska, Katarzyna Tajchman, Piotr Domaradzki, Mariusz Florek

**Affiliations:** 1Institute of Parasitology of the Polish Academy of Sciences, Research Station in Kosewo Górne, 11-700 Mrągowo, Poland; kosewopan@kosewopan.pl; 2Department of Animal Ethology and Wildlife Management, Faculty of Animal Sciences and Bioeconomy, University of Life Sciences in Lublin, Akademicka 13, 20-950 Lublin, Poland; 3Department of Quality Assessment and Processing of Animal Products, University of Life Sciences in Lublin, Akademicka 13, 20-950 Lublin, Poland; piotr.domaradzki@up.lublin.pl

**Keywords:** *Dama dama*, bone marrow, fatty acid, pasture period, winter nutrition

## Abstract

**Simple Summary:**

The bone marrow is regarded as an indicator of the health status in ungulates, because it is not only a source of fat but also it contains minerals, vitamins, and fatty acids. These substances affect the functioning of the organism, especially in unfavorable conditions such as winter. However, the relevant concentrations of fatty acids for healthy deer bone marrow are not known; their contents can be modified by various nutrition, although for ruminants there are some limitations. Unfortunately, studies on the composition of bone marrow fatty acids or their changes are scarce, especially in relation to the youngest deer. Therefore, the aim of the study was to analyze the fatty acid composition of the bone marrow of farmed fallow deer fawns after the pasture period and the winter feeding. The results obtained in the study indicate that, notwithstanding inclement weather conditions in winter, providing the animals with appropriate housing conditions on the farm as well as their optimal feeding resulted in better condition and nutritional status of the animals, which is indicated by a more favourable composition and profile of fatty acids in bone marrow, compared to the optimal summer grazing period for ruminants.

**Abstract:**

There are few studies on the composition of fatty acids and how they change the bone marrow fat of young animals depending on nutrition. Therefore, the proximate and fatty acid composition of metatarsal bone marrow from fawns of farm fallow deer after a summer of grazing and the winter feeding was compared. Due to the size and nature of the data, parametric or nonparametric tests were used. Fatty acid composition was determined by gas chromatographic analysis. After the winter feeding, bone marrow contained more fat (83.11% vs. 75.09%, *p* < 0.05) and less fat free dry matter (5.61% vs. 13.76%, *p* < 0.05) compared to the pasture period. Moreover, there was a significantly higher amount of saturated fatty acids (23.34% vs. 21.60%, *p* < 0.001), more *trans* fatty acids (2.99% vs. 2.34%, *p* < 0.005), and conjugated linoleic acid isomers (1.04% vs. 0.83%, *p* < 0.01), compared to post winter feeding, which in turn contained significantly more total *cis*-monounsaturated fatty acids (54.65% vs. 58.90%, *p* < 0.001). The percentage of polyunsaturated fatty acids (including *n*-3 and *n*-6) was not affected by feeding season. In conclusion, it was shown that young male farm fallow deer were better nourished after the winter period, during which they were kept in properly prepared rooms and fed fodder prepared by people.

## 1. Introduction

Deer have a unique biology linked to their characteristic feature: the antler. As with many other secondary sexual traits of males, antler size strongly depends on external factors, particularly nutrition. Cervids are a group of mammals with relatively high nutritional needs due to the important demands of their life strategies [1,2,3]. When assessing the body condition of deer, it is recommended that the percentage of fat in the bone marrow be determined, and most studies follow this suggestion [4]. However, the marrow of young animals is not only a source of fat but also a site of various concentrations of minerals, vitamins and fatty acids that affect the functioning of the young animal, especially during unfavourable periods [5]. In bone marrow, cells are multiplied and circulated through the bloodstream for growth and remodelling [6]. The optimal concentrations of fatty acids and other compounds for healthy deer bone marrow are not known; however, it seems probable that it is beneficial to have sufficient concentrations to move around efficiently and maintain an adequate supply of backup tissue for winter [7,8]. Animals take up nutrients and structural components from the gastrointestinal tract transported by the circulatory system according to their needs and can alter absorption efficiency to meet changing requirements [9]. Young animals have been shown to absorb more Ca^+2^, for example, compared to mature ones [10]. Quantitatively, the development and maintenance of the skeletal system is the most important function of minerals, and their accumulation in the skeletal tissue is interdependent: one element will not accumulate without the other, and a synergy exists among certain vitamins (especially the fat-soluble vitamin A and D) and fatty acids (especially the necessary unsaturated fatty acids) [11,12].

The fatty acid (FA) content can be modified by various nutritional additives, and this issue has been relatively well investigated in livestock. The composition of FAs of animal products (meat, milk, and eggs) reflects both the tissue biosynthesis of fatty acids and the fatty-acid profile of lipids supplied with nutrition [13,14,15,16]. This relationship has been shown to be stronger in monogastric animals than in ruminants, where dietary fatty acids are hydrogenated in the rumen [17,18]. For these reasons, in monogastric animals, PUFAs are usually present in adipose tissue in large amounts [19]. This group of acids is probably mobilized from a plant-based diet, as they cannot be synthesized in the body [20]. It has also been shown that the trans fatty acids present in the lipids of all ruminants are the result of the bacterial metabolism of PUFAs in their rumen. The natural content of these acids in ruminant fat generally does not exceed 6% [21].

Bone marrow fat content has long been associated with the physiological condition of animals [22,23,24,25]. Selective deposition of unsaturated fatty acids in the bone marrow may facilitate the mobilization of these fatty acids at low temperatures. It has been shown that mainly long-chain and unsaturated fatty acids are released from adipose tissue during lipolysis [26,27].

Many researchers regard bone marrow as an indicator of the health status in ungulates, as it is the last adipose tissue to be mobilized during starvation periods [23,28,29,30]. Unfortunately, studies on the composition of bone marrow fatty acids or their changes are scarce, especially in relation to the youngest animals and their nutritional status. It is important for fawns to reach an appropriate body weight before winter as the condition of their bodies before adulthood affects their survival not only over the following few months but also in subsequent years [31,32,33]. However, achievement of sufficient body weight does not necessarily reflect a favourable fatty acid composition in the bone marrow, as demonstrated in our study on red deer [34]. In neonatal and suckling animals, a large part of the marrow is occupied by its red component where hematopoietic cells are formed. Prenatal, neonatal, and juvenile ruminants are characterized by low levels of linoleic and linolenic acid in tissue lipids [35,36,37]. On this basis, it has been suggested that the youngest animals may be on the verge of essential fatty acid deficiency [38]. Moreover, the link between diet and body condition is highly relevant for managing large herbivores, especially in fenced ecosystems with limited resources [39].

Information on the nutritional management of cervids is still limited, meaning that production is still not optimized for these species, which means that investment strategies cannot be studied under perfect nutritional conditions [40,41]. Therefore, the aim of the study was to analyse the fatty-acid composition of the bone marrow of farmed fallow deer fawns after the pasture period and after the winter feeding.

## 2. Materials and Methods

### 2.1. Experimental Design

The studies were carried out on 13 male farmed fallow deer (*Dama dama*) in the first year of life; i.e., six 6–7-month-old animals after the grazing period (November) and seven 11–12-month-old male after the winter period (April). Young stags are usually culled in the breeding process, while hinds are intended for reconstruction of the herd. The animals were reared at the Research Station of the Institute of Parasitology, Polish Academy of Sciences (Region of Warmia and Mazury; Kosewo Górne, Poland; N: 53°48′; E: 21°23′), under constant observation and emergency veterinary care. The breeding system was based on rotational pasture. The area and density of the plots were consistent with the recommendations proposed by Department for Environment Food and Rural Affairs (DEFRA) [42], Federation of European Deer Farmers Associations (FEDFA) [43], and Mattiello [44]. The study involved fawns born in a natural way during the grazing period, which lasts from April to November in Poland. At the beginning of their lives, they were fed with milk by the doe, and later they ate the vegetation available on the pasture. In the winter period (from December to March), the animals were fed ad libitum with grass haylage or hay with a moderate nutritional value. Each animal ingested on average 260 g d^−1^ of a mixture comprising 70% crushed oats, 15% rapeseed concentrate (containing 33% crude protein; Eko-pasz, Mońki, Poland), and 15% of soybean concentrate (with 45% crude protein content; Eko-pasz, Mońki, Poland) and Josera Phosphoreimer multi-ingredient licks (Josera, Nowy Tomyśl, Poland).

### 2.2. Sampling

The body weight of the farmed animals was measured before slaughter using MP 800 sensors coupled with a Tru-test DR 3000 weight reader (accuracy: ±1%, minimum resolution: 100 g). The weight of the fallow deer was estimated from the carcass weight, which accounts for 67% of the body weight of these animals [45]. The carcass weight was determined after culling and evisceration of the animals.

Bones and bone marrow were sampled in November 2019 and in April 2020. The samples were collected from the farmed animals during routine slaughter, which is the final stage of breeding. On the day of slaughter, metatarsal bone (*ossa metatarsalia*) samples were collected from each animal. The metatarsal bone was dissected by separating the skin, muscles, and tendons with a stainless-steel knife. Fresh bones were opened with a dental titanium drill carefully to avoid contamination of the bone marrow, which was collected after the removal of the red part and frozen (−20 °C).

### 2.3. Determination of Fat, Moisture, and Fat-Free Dry Mass in Bone Marrow

A bone marrow sample weighing approximately 1.5 g (accuracy up to 1 mg) was placed in a preweighed cellulose thimble. The percent moisture loss was determined according to PN-ISO 1442:2000 [46] with the drying method (103 °C) using a universal oven Memmert UF30 (Schwabach, Germany). The bone marrow fat was extracted with the Soxhlet lipid extraction method using Büchi-B-811 (Flawil, Switzerland) equipment and n-hexane as a solvent according to PN-ISO 1444:2000 [47], and finally the percentage of fat was calculated. The cellulose thimble with the remaining fat-free dry matter was dried at approximately 103 °C for 1 h and then cooled to room temperature in a glass desiccator; next, the percentage of dry matter was calculated.

### 2.4. Fatty Acid Analysis in Bone Marrow

The fatty acid (FA) profile in the bone marrow was determined after fat extraction according to the method proposed by Folch et al. [48] method. FA methyl esters (FAMEs) were prepared via transmethylation of the fat samples (50 mg) using a mixture of concentrated H_2_SO_4_ (95%) and methanol according to the AOCS Official Method Ce 2-66 [49]. Gas chromatographic (GC) analyses were performed according to Domaradzki et al. [50] using a Varian CG 3900 (Walnut Creek, CA, USA) gas chromatograph with a flame ionization detector (FID). The FAMEs were separated in a CP 7420 capillary column (Agilent Technologies, Santa Clara, CA, USA; 100 m in length, inner diameter 0.25 mm, film thickness 0.25 μm). The analysis was carried out in increasing temperature conditions. The temperature program was as follows: 50 °C for 1 min, 30 °C/min up to 120 °C, 2 °C/min up to 160 °C, 30 min holding, 1 °C/min up to 200 °C, 5 °C/min up 250 °C, and 1 min holding time. The temperature of the injector and the detector was 260 °C and 270 °C, respectively; the carrier gas (hydrogen) had a flow rate of 2 mL/min, the volume of the injected samples was 1 μL, and the split ratio was 1:50. The identification and quantification of FAMEs were based on retention times corresponding to reference mixtures (Supelco 37 Component FAME Mix CRM 47885 Supelco Inc., Bellefonte, PA, USA; CLA methyl ester O5632—Sigma-Aldrich, St. Louis, MO, USA; Branched Chain FAME Mixture BR2, BR3, BR4—Larodan AB Solna, Sweden). Star GC Workstation v5.5. software (Varian Inc., Walnut Creek, CA, USA) was used. The fatty acid composition was expressed as a percentage of total identified FAs. The analyses were carried out in duplicate.

The FAs were recorded using numerical symbols containing the number of carbon atoms in the FA chain (the number before the colon), the number of double bonds (the number after the colon), and the double bond geometry (*cis* or *trans*). The notation “*n*” was used for FAs representing the *n*-3, *n*-6, and *n*-9 family, where the *n*-number indicates the position of the first double bond counted from the methyl terminal end of the carbon chain. The following groups of fatty acids were identified and their ratios and indices were calculated: SFA—saturated FAs, even-numbered; OCFA—odd-chain FAs; BCFA—branched-chain FAs; MUFA *cis*—monounsaturated FAs, even-numbered; PUFA—polyunsaturated FAs; *n*-3 and *n*-6 FAs; TFA—*trans* FAs: sum of MUFA *trans*, ∑18:2 *trans* and ∑C18:3 *trans*; ∑CLA—conjugated linoleic acid: sum of 18:2 *c*9,*t*11, 18:2 *t*9,*c*11; ∑C18:2 *trans*—sum of non-conjugated 18:2 *t*,*c*/ *c*,*t*/ *t*,*t* isomers; ∑C18:3 *trans*—sum of 18:3 *trans* isomers with an unknown position of double bonds, and the PUFA/SFA, MUFA/SFA, and *n*-6/*n*-3 ratios.

### 2.5. Statistical Analysis

The statistical analyses were performed in Statistica v13 (TIBCO Software Inc., Palo Alto, CA, USA). The statistical differences in the individual variables between the fallow deer groups were analysed depending on the results of the Shapiro–Wilk test of normality. Variables normally distributed were tested using the unpaired Student’s *t*-test (for two independent groups), otherwise the Kolmogorov–Smirnov test was applied. The results were expressed as the mean value and standard deviation of the variables. A principal component analysis (PCA) was further applied to visualize data and demonstrate the relationships between variables characterizing the fatty acid profile and composition in the bone marrow and the fallow deer diet.

## 3. Results

The mean marrow concentrations of fat, moisture, fat-free dry matter, and fatty acids were compared between feeding seasons (Table 1). Clearly, the body weight of the animals differed significantly (*p* < 0.001), which was related to different ages (autumn, 6–7 months; spring, 11–12 months) (Table 1). Significant differences were also found in the content of fat (F) and fat-free dry matter (FFM) (*p* < 0.05), depending on the season. The higher mean fat content was determined after the winter nutrition period (April) rather than after the pasture period (November). An opposite relationship was revealed in the FFM levels, which was higher in the bone marrow after the pasture period (Table 1).

The gas chromatography analysis (GC–FID) allowed for the effective separation and identification of 47 FAs in the bone marrow fat (BMF), taking into account their configuration. Trace amounts of the following fatty-acid methyl esters (FAMEs) were detected: C10:0, C12:0, C13:0, C20:0, C21:0, C16:1 *c*11, C17:1 *c*7, C20:1 *c*13, C20:2 *n*-6, C20:5 *n*-3 EPA, and C22:4 *n*-6; moreover, in the case of the next 22 FAs, their unit share did not exceed 1%. Hence, Table 2 presents 14 selected FAs, the proportion of which in the bone marrow samples was not less than 1%.

In the BMF after the pasture period, compared to deer after the winter feeding, significantly more ΣSFAs (*p* < 0.001) were found, which was a consequence of a significantly higher share of C16:0 (*p* < 0.001) and C18:0 (*p* < 0.05), as well as higher ΣTFA (*p* < 0.005), ΣMUFA *trans* (*p* < 0.005), ΣC18: 2 *trans* (*p* < 0.001), and ΣCLA (*p* < 0.01). In turn, the bone marrow of fallow deer after the winter nutrition showed a higher share of ΣMUFA *cis* (*p* < 0.001), including C16:1 *c*9 (*p* < 0.05), C17:1 *c*9 (*p* < 0.05), and C18:1 *c*11 (*p* < 0.001, Table 1).

The mean share of oleic acid responsible for the fluidity of the bone marrow in the distal parts of deer legs did not differ significantly between the feeding seasons, and there was a lower content determined after the grazing season (*p* = 0.070).

It is worth noting that although no significant differences in total PUFAs were found, a higher level including Σ*n*-3 and a more favourable *n*-6/*n*-3 ratio were observed in in the BMF after pasture feeding. The PUFA/SFA ratio in both groups was the same, which was mainly related to the higher proportion of saturated fatty acids in the BMF of fallow deer fed on pasture (Table 1). The principal component analysis (PCA) was performed for a more extended analysis of the relationships between the 17 variables characterizing the bone marrow and the body weight of the animals as an additional variable. Three principal components (PCs) with eigenvalues exceeding 1 (Kaiser criterion) explained 89.95% of the total variance, with PC1, PC2, and PC3 accounting for 50.34%, 31.88%, and 7.73% of the variance, respectively (Table 2).

Table 3 shows the correlation coefficients between the PCs and variables obtained from the correlation matrix. The first component (PC1) exhibits a very high positive correlation with ΣMUFA *cis* (r = 0.922), C18:1 *c*11 (r = 0.919), body weight (r = 0.847), and C16:1 *c*9 (r = 0.764), but a high negative correlation with six variables, namely, ΣC18:2 *trans* (r = −0.919), TFA (r = −0.913), ΣMUFA *trans* (r= −0.882), C18:1 *t*11 VA (r = −0.875), C16:0 (r = −0.830), and SFA (r = −0.829) (Table 3). In turn, the second component (PC2) is characterized by a high positive correlation with 4 variables describing the polyunsaturated fatty acids, namely, PUFA, *n*-3, *n*-6, and the PUFA/SFA ratio (0.836 ≤ r ≤ 0.885), and a negative correlation with C15:1 (r = −0.755). The third component (PC3) displays the closest positive correlation with SFA (r = 0.471) and C16:0 (r = 0.412) and a negative relationship with PUFA/SFA (r = −0.442).

Taking into account the loadings and the length of the directional vectors shown in Figure 1 (as a two-factor plane PC1 × PC2), three apparent groups of parameters can be distinguished. The first group of variables, which included monounsaturated fatty acids with the *cis* conformation (ΣMUFA *cis*, C18:1 *c*11, and C16:1 *c*9) and additionally the body weight, was distributed in the positive area of the first PC (Q1 and Q2, upper and bottom right quadrants). The second group comprised saturated fatty acids (SFA and C16:0) and trans fatty acids (TFA, ΣMUFA *trans*, C18:1 *t*11 VA, and ΣC18:2 *trans*), which were located in the positive area of PC2 (Q3 and Q4, upper and bottom left quadrants). The third group comprising PUFA, *n*-3, *n*-6, and PUFA/SFA was located in the upper right quadrant (Q4) defined by positive values of PC2 and negative values of PC1.

Figure 2 shows the projection of cases depending on the fallow deer diet in the coordinate system defined by PC1 × PC2.

The samples of the bone marrow from the animals fed indoors (in the winter period) are clearly separated and situated in the right area of the plot; i.e., they have positive values of PC1. This area represents the highest values for ΣMUFA *cis*, C18:1 *c*11, C16:1 *c*9, and body weight (Figure 1), which clearly corresponds to the results shown in Table 1 for this fallow deer group. The second group, in turn, is composed of bone marrow samples from the pastured animals, which were distributed on the left area of the plot, with negative values of PC1. This area is distinctive for saturated and trans fatty acids, e.g., SFA, C16:0, TFA, ΣMUFA *trans*, C18:1 *t*11 VA, and ΣC18:2 *trans* (Figure 1), for which significantly higher values were reported in the case of the bone marrow from the grazing animals. The third group of variables, comprising polyunsaturated acids, located in the Q4 square of Figure 1, appears to be neutral due to the absence of significant differences related to the fallow deer diet.

## 4. Discussion

The results of this research on young male farm fallow deer showed that the bone marrow FA composition changed with age, which is directly related to the feeding season according to the established production cycle on the farm. In addition, the fawns, after pasture feeding, showed both a lower fat concentration and a higher FFM, which may indicate malnutrition before the winter period [28,51,52]. On the one hand, it can be assumed that the young males were better nourished after the winter period, which would confirm the practice of wintering the fawns because those aged 6–7 months (November) do not have sufficient reserves fatty acids in their bones [53]. On the other hand, twice the amount of fat-free mass after the grazing season indicates a greater amount of red bone marrow compared to yellow. It has been shown that in young animals, the red fraction may dominate [45]. The proportion of both bone fractions significantly changes in the spring in older animals because the amount of adipose tissue increases, while the FFM decreases.

In the study by Soppel and Nieminen [25], the marrow in the metatarsal bones of reindeer calves showed more than 50% more ΣSFA, including palmitic acid (C16:0) and stearic acid (C18:0) and fewer MUFAs (average 33.88 vs. 56.78%) compared with the BMF of fawns assessed in both feeding seasons. In the case of ∑PUFAs and ∑*n*-3, the BMF values were similar among the animals; however, the bone marrow of reindeer calves contained more PUFAs from the *n*-6 group, especially linoleic acid (C18:2 *n*-6; average 1.92 vs. 1.45%). The differences could have resulted from interspecies differences, diet or age: the reindeer were younger than the fallow deer.

Conjugated linoleic acid (CLA) isomers were found in the marrow of antelope (1.5%), elk (1.0%) and deer (1.0%) [54]. In studies on farm fallow deer fawns, the bone marrow content of this acid was the same or lower (around 0.93%). A significantly higher share of ∑CLAs was found in the bone marrow of fawns that nursed during the pasture season compared to those fed in winter. This was likely associated with longer milk-feeding from a better-fed doe because ruminant milk may contain high levels of CLA [55].

In ruminants, PUFAs are present in adipose tissue in large amounts due to their efficient hydrogenation by rumen microorganisms [19], which was confirmed by the results obtained from the farmed fallow deer. The PUFA content in the metatarsal bone marrow was either equal to or higher than that for the malnourished reindeer, which probably indicated a high mobilization of these fatty acids from the plant diet, especially after the pasture period due to their exogenous nature [20].

The composition of fatty acids in the bone marrow may have varied depending on the bones: the more distal ones were dominated by unsaturated fatty acids, which lowered the melting point of fat [56]. Sugár and Nagy [56] in the metatarsal bones of fallow deer showed a decrease in the share of palmitic and stearic acids to 14% and 2.5%, respectively; at the same time, they observed an increase in the share of palmitoleic and oleic acids to 21% and 43%, respectively. Compared to the results of Sugár and Nagy [56], the authors of this research found the share of C16:0 and C18:0 acids in the marrow of farm fallow deer to be higher, especially in the pre-winter period, and the share of palmitoleic and oleic acids to be lower. The inverse relationship (i.e., a lower proportion of C16:0 and C18:0 acids in the bone marrow and a greater proportion of palmitoleic and oleic acids) was observed in the spring.

In the fallow deer bone marrow, several times more MUFAs than PUFAs were found, both in November and April. It should be emphasized that the dominant acid in the bone marrow was oleic acid, which accounted for nearly 40% of the sum of the FAs. The high proportion of oleic acid determined the low melting point of fat in the bone marrow, which increased its fluidity and thus the functioning of the limbs during a frost [7,8]. The mechanism of maintaining the softness of fats by increasing the degree of their unsaturation has been observed in the distal bones or limbs of various species [8,57,58,59]. Irving et al. [7,60] determined that changes in bone marrow melting temperature were not limited to arctic mammals, but also occurred in tropical mammals.

## 5. Conclusions

Based on the bone marrow studies of young male farm fallow deer, those in the winter had a more favourable nutritional status, which resulted from properly provided living conditions and properly balanced feed. The bone marrow of fawns after the winter feeding, contained fewer saturated and trans fatty acids, and more monounsaturated FAs, with a similar level of PUFAs than after pasture feeding. It can therefore be assumed that supplementing the fallow deer diet with unsaturated fatty acids before the winter period favourably modifies the fatty acid composition in the bone marrow. However, further studies are needed to confirm this hypothesis. In addition, learning about the composition of fatty acids in the bone marrow of farm animals will allow the level of animal nutrition to be determined, which leads to appropriate supplementation at the appropriate stage of rearing.

## Figures and Tables

**Figure 1 animals-12-00941-f001:**
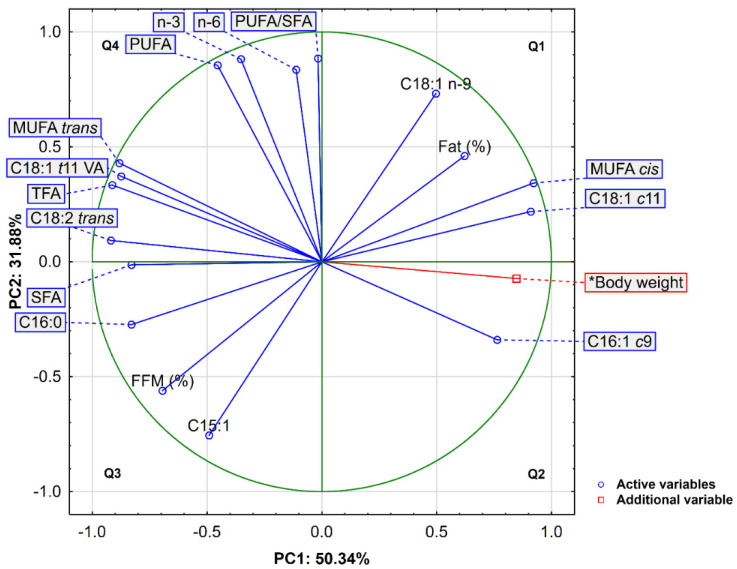
Projection of variables in a two-factor plane (PC1 × PC2); Fat (%)—fat content; TFA—*trans* fatty acids; FFM (%)—fat-free dry matter.

**Figure 2 animals-12-00941-f002:**
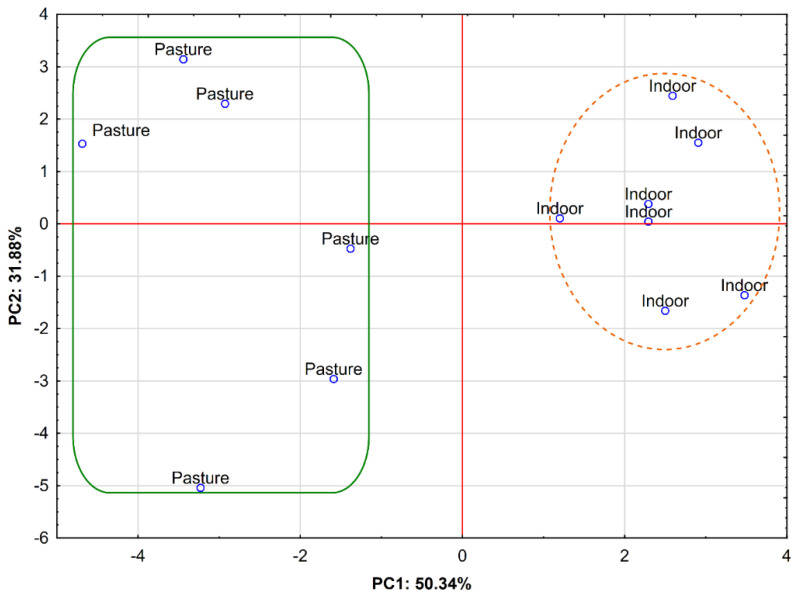
Projection of cases depending on the fallow deer diet (Pasture vs. Indoor) in the a two-factor plane (PC1 × PC2).

**Table 1 animals-12-00941-t001:** Body weight, composition, and fatty acid percentages of bone marrow of fallow deer, depending on diet.

Variable	Pasture		Indoor		Statistics	
	M	SD	M	SD	*p*-Value	Test
Body weight (kg)	32.17	3.10	41.57	4.31	0.001	t
Bone marrow composition (%)						
Fat	75.09	6.60	83.11	5.32	0.034	t
Moisture	11.15	1.53	11.24	1.45	0.914	t
FFM	13.76	6.48	5.61	3.69	0.016	t
Fatty acid percentage (%)						
C14:0	2.59	0.34	2.68	0.37	0.637	t
C16:0	17.44	0.58	15.95	0.53	<0.001	t
C18:0	3.18	0.32	2.80	0.26	0.039	K-S
∑SFA	23.34	0.87	21.60	0.61	0.001	t
C15:0	1.06	0.07	0.98	0.09	0.140	t
∑OCFA	11.42	1.80	10.03	0.61	0.079	t
C18:0*iso*	1.33	0.23	1.32	0.21	0.938	t
∑BCFA	3.70	0.39	3.48	0.18	>0.100	K-S
C14:1 *c*9	1.72	0.18	1.96	0.27	0.101	t
C15:1	8.14	1.70	6.67	0.51	0.052	t
C16:1 *c*9	11.84	0.80	13.26	0.89	0.012	t
C17:1 *c*9	1.45	0.10	1.61	0.10	0.020	t
C18:1 *n-*9	37.74	1.67	39.14	0.76	0.070	t
C18:1*c*11	1.77	0.12	2.75	0.26	< 0.001	t
∑MUFA *cis*	54.65	1.33	58.90	0.95	<0.001	t
C18:2 *n*-6 LA	1.44	0.11	1.46	0.09	0.776	t
CLA	1.04	0.13	0.83	0.10	0.008	t
∑PUFA	3.88	0.43	3.63	0.29	0.243	t
∑*n-3*	0.98	0.18	0.92	0.10	>0.100	K-S
∑*n-6*	1.65	0.15	1.66	0.13	0.831	t
∑MUFA *trans*	1.60	0.20	1.21	0.08	<0.005	K-S
ΣC18:2 *trans*	1.07	0.12	0.77	0.05	<0.001	t
∑TFA	2.99	0.34	2.34	0.07	<0.005	K-S
*n-6/n-3*	1.71	0.21	1.81	0.18	0.379	t
PUFA/SFA	0.17	0.02	0.17	0.02	0.834	t

M—mean; SD—standard deviation; K-S—Kolmogorov–Smirnov test; t—Student’s *t*-test; FFM—fat-free dry mass; SFA—sum of saturated fatty acids; OCFA—odd-chain fatty acids; BCFA—branched-chain fatty acids; MUFA *cis*—sum of *cis* monounsaturated fatty acids; LA—linoleic acid 18:2 *n*-6; PUFA—sum of polyunsaturated fatty acids; MUFA *trans*—sum of trans monounsaturated fatty acids; TFA—sum of *trans* fatty acids; CLA—sum of conjugated linoleic acid isomers.

**Table 2 animals-12-00941-t002:** Eigenvalues and the proportion of variation (%) explained by the principal components.

Component	Eigenvalue	Proportion	Cumulative
1	8.56	50.34	50.34
2	5.42	31.88	82.22
3	1.31	7.73	89.95
4	0.63	3.72	93.67
5	0.40	2.34	96.01
6	0.29	1.73	97.74
7	0.17	1.02	98.76
8	0.13	0.76	99.52
9	0.05	0.30	99.82
10	0.02	0.12	99.94
11	0.01	0.06	99.99
12	0.00	0.00	100.00

**Table 3 animals-12-00941-t003:** Correlations between the principal components and the original variables.

Variable	PC1	PC2	PC3
Body weight (kg)	0.847	−0.073	0.398
Fat (%)	0.621	0.460	0.264
FFM (%)	−0.695	−0.560	−0.252
SFA	−0.829	−0.013	0.471
C16:0	−0.830	−0.273	0.412
MUFA *cis*	0.922	0.343	0.019
C16:1 *c*9	0.764	−0.339	−0.402
C18:1 *c*11	0.910	0.218	0.142
C18:1 *n*-9	0.496	0.732	0.339
C15:1	−0.491	−0.755	−0.326
MUFA *trans*	−0.882	0.429	0.035
C18:1 *t*11 VA	−0.875	0.372	0.119
C18:2 *trans*	−0.919	0.093	−0.191
TFA	−0.913	0.334	−0.063
PUFA	−0.454	0.855	−0.156
*n*-3	−0.353	0.882	−0.038
*n*-6	−0.112	0.836	−0.330
PUFA/SFA	−0.017	0.885	−0.442

## Data Availability

Data sharing not applicable.

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
