# Peer review of "Composition and Fatty Acid Profile of Bone Marrow in Farmed Fallow Deer (Dama dama) Depending on Diet"

_animals, 2022, doi:10.3390/ani12080941_

Round 1
Reviewer 1 Report
Dear Author,
You have done an excellent work that included hard work and patience.
1) Please reframe the sentence in 50th line.
2) Please correct the Ca to Ca+2
3) In 65th line insert - and
4) After the winter feeding - add the in manuscript to make it parallel
5) Provide two full forms in108th line
6) Statistical section needs to rewritten properly. Currently, it is not clear completely, in t-test, please mention which type you used.
7)
| the results of the Shapiro-Wilk test of | 177 |
| normality using the t-test (together with Levene’s test) or the Kolmogorov-Smirnov test. |
After Levene's test is over, we go t -test
8) Please correct representation in line 201
Author Response
Dear Reviewer,
The authors would like to warmly thank you for all comments and suggestions, especially the critical ones, aimed at improving the scientific value of the article and eliminating the most important errors. We would like to inform that all Reviewer’s comments and corrections were introduced in amended manuscript.
Authors are greatly appreciate the opportunity that have been given to further revise the manuscript, and believe that Reviewers will share submitted arguments and find this revision fully satisfactory.
Answers and explanations to all the issues raised are given below.
Reviewer #1
Dear Author,
You have done an excellent work that included hard work and patience.
Answ. Thank you kindly for this positive and very motivating comment.
1) Please reframe the sentence in 50th line.
Answ. Thank you very much for valuable comment. This sentence was redrafted as suggested.
2) Please correct the Ca to Ca+2
Answ. Thank you this remark, this error was corrected.
3) In 65th line insert - and
Answ. Thank you this remark, ‘and’ was inserted where appropriate.
4) After the winter feeding - add the in manuscript to make it parallel
Answ. Thank you this remark, the manuscript was unified.
5) Provide two full forms in108th line
Answ. Thank you this remark. The acronyms have been explained
6) Statistical section needs to rewritten properly. Currently, it is not clear completely, in t-test, please mention which type you used.
7) the results of the Shapiro-Wilk test of 177
normality using the t-test (together with Levene’s test) or the Kolmogorov-Smirnov test.
After Levene's test is over, we go t -test
Answ. Thank you very much for valuable comment. The statistical section was corrected.
8) Please correct representation in line 201
Answ. Thank you very much for valuable comment. The table caption has been corrected.
Thank you for your positive and substantive review.
Yours faithfully,
Mariusz Florek, Professor
Corresponding author
Reviewer 2 Report
Dear authors,
I revised the manuscript identified as animals-1647877. In my opinion the manuscript is interesting. The methods used are described in detail and consistent with the aims of the research. The results are well presented and derive from the research conducted. The discussions are well focused on results. However, some changes or implementations would need to be made. Authors are requested to check as the chemical formulas are reported (for example at line 144). In addition, I suggest verifying that in the tables note there are omissions (for example to table 1, line 205, where the indication of the value of "p" is missing). Finally, I believe that the authors can improve the introduction, in particular from lines 63 to 68, by also referring to the effect of seasonal variations on the quality of the pasture. In this regard, the authors may refer to the following manuscript (https://doi.org/10.3390/foods9081091), which I strongly encourage you to use as a reference.
I wish you good luck
Sincerely
Author Response
Dear Reviewer,
The authors would like to warmly thank you for all comments and suggestions, especially the critical ones, aimed at improving the scientific value of the article and eliminating the most important errors. We would like to inform that all Reviewer’s comments and corrections were introduced in amended manuscript.
Authors are greatly appreciate the opportunity that have been given to further revise the manuscript, and believe that Reviewers will share submitted arguments and find this revision fully satisfactory.
Answers and explanations to all the issues raised are given below.
Reviewer #2
Dear authors,
I revised the manuscript identified as animals-1647877. In my opinion the manuscript is interesting. The methods used are described in detail and consistent with the aims of the research. The results are well presented and derive from the research conducted. The discussions are well focused on results. However, some changes or implementations would need to be made. Authors are requested to check as the chemical formulas are reported (for example at line 144). In addition, I suggest verifying that in the tables note there are omissions (for example to table 1, line 205, where the indication of the value of "p" is missing). Finally, I believe that the authors can improve the introduction, in particular from lines 63 to 68, by also referring to the effect of seasonal variations on the quality of the pasture. In this regard, the authors may refer to the following manuscript (https://doi.org/10.3390/foods9081091), which I strongly encourage you to use as a reference.
I wish you good luck
Sincerely
Answ. Thank you kindly for this favourable comment and all noted suggestions. We would like to inform that chemical formula was corrected. In the case of p value (L205) our intention was slightly different, as a specific p-value was to be indicated by the level given in the table. Therefore, this information under the table was dropped to avoid confusion. Finally, suggested reference has been included in the Introduction and References [16]. Thank you very much for positive and substantive review.
Thank you for your positive and substantive review.
Yours faithfully,
Mariusz Florek, Professor
Corresponding author

Reviewer 3 Report
The title did not accurately reflect the major findings of the survey. I suggest to Authors to chnge it by specifying the studied ungulates.
The abstract adequately summarizes methodology, results, and significance of the study. However, Authors should indicate the statistical analysis.
Keywords represent the article adequately.
The introduction section is well written and it falls within the topic of the study.
The section of Material and Methods is clear for the reader and well describes the methods applied in the study. How the health status of animals has been assessed? Generally, an animal is considered healthy when showed normal physiologic parameters following a clinical examination (heart rate, respiratory rate, and rectal temperature), and haematochemical indices following routine hematology and plasma biochemistry testing.
This aspect need to be clarified.
Regarding statistical analysis, Authors wrote “The statistical differences in the individual variables between the fallow deer groups were analyzed depending on the results of the Shapiro-Wilk test of normality using the t-test (together with Levene’s test) or the Kolmogorov-Smirnov test”
Authors should indicate what data passed normality test and the statistical model applied, consequently.
Results section as well as Discussion section is well written. The findings obtained in the study were well discussed and justified with appropriate references.
The conclusion paragraph well summarizes the results and the significance of the study.
However, I suggest to change “…this thesis requires further detailed research…” with “…further studies are needed to confirm this hypothesis…”
The tables and Figures are generally good and well represent the results of the study.
Authors should check and standardize the references in the list according to journal guidelines.
Author Response
Dear Reviewer,
The authors would like to warmly thank you for all comments and suggestions, especially the critical ones, aimed at improving the scientific value of the article and eliminating the most important errors. We would like to inform that all Reviewer’s comments and corrections were introduced in amended manuscript.
Authors are greatly appreciate the opportunity that have been given to further revise the manuscript, and believe that Reviewers will share submitted arguments and find this revision fully satisfactory.
Answers and explanations to all the issues raised are given below.
Reviewer #3
The title did not accurately reflect the major findings of the survey. I suggest to Authors to change it by specifying the studied ungulates.
Answ. Thank you for this comment, although we thought the title was sufficiently precise. Nevertheless, the Latin name for fallow deer is also included in the title.
The abstract adequately summarizes methodology, results, and significance of the study. However, Authors should indicate the statistical analysis.
Answ. Thank you very much for valuable comment. The information about statistical analysis was introduced as suggested.
Keywords represent the article adequately.
Answ. Thank you for this comment.
The introduction section is well written and it falls within the topic of the study.
Answ. Thank you for this comment.
The section of Material and Methods is clear for the reader and well describes the methods applied in the study. How the health status of animals has been assessed? Generally, an animal is considered healthy when showed normal physiologic parameters following a clinical examination (heart rate, respiratory rate, and rectal temperature), and haematochemical indices following routine hematology and plasma biochemistry testing.
This aspect need to be clarified.
Answ. As stated in the Material and methods chapter all fallow deer came from the Research Station of the Institute of Parasitology (Polish Academy of Sciences, http://ipar.pan.pl/en/research-stations/). The animals were provided with the level of welfare appropriate to the environmental conditions, as well as permanent observation and emergency veterinary care related to e.g. cases of disease or standard anthelmintic therapy. However, it should be borne in mind that these are farmed wild animals for which the standard treatment as for domestic animals cannot be fully utilised.
We would like also to inform that the relevant information was additionally included in the Experimental design section.
Regarding statistical analysis, Authors wrote “The statistical differences in the individual variables between the fallow deer groups were analyzed depending on the results of the Shapiro-Wilk test of normality using the t-test (together with Levene’s test) or the Kolmogorov-Smirnov test”
Authors should indicate what data passed normality test and the statistical model applied, consequently.
Answ. Thank you for the important comment. We would like to inform that Statistical analysis section was redrafted and clarifying information was included. We thus hope that the description of Table 1 is sufficiently clear to readers and accurately states which variables had a normal distribution (Student's t-test) and for which such a distribution was not found (Kolmogorov-Smirnov test).
Results section as well as Discussion section is well written. The findings obtained in the study were well discussed and justified with appropriate references.
Answ. Thank you very much for this kind comment.
The conclusion paragraph well summarizes the results and the significance of the study.
Answ. Thank you very much for this comment.
However, I suggest to change “…this thesis requires further detailed research…” with “…further studies are needed to confirm this hypothesis…”
Answ. Thank you very much for valuable comment. This sentence was corrected as suggested.
The tables and Figures are generally good and well represent the results of the study.
Answ. Thank you very much for this comment.
Authors should check and standardize the references in the list according to journal guidelines.
Answ. Thank you for this comment. All references have been checked.
Thank you for your positive and substantive review.
Yours faithfully,
Mariusz Florek, Professor
Corresponding author
